# Bird species define the relationship between West Nile viremia and infectiousness to *Culex pipiens* mosquitoes

**Jefferson A. Vaughan**[1]*, **Robert A. Newman**[1], **Michael J. Turell**[2¤]

**1** Department of Biology, University of North Dakota, Grand Forks, North Dakota, United States of America,
**2** Virology Division, US Army Medical Research Institute of Infectious Diseases, Fort Detrick, Maryland, United States of America

¤ Current address: VectorID LLC, Frederick, Maryland, United States of America
* jefferson.vaughan@und.edu

## Abstract

The transmission cycle of West Nile virus (WNV) involves multiple species of birds. The relative importance of various bird species to the overall transmission is often inferred from the level and duration of viremia that they experience upon infection. Reports utilizing *in vitro* feeding techniques suggest that the source and condition of blood in which arboviruses are fed to mosquitoes can significantly alter the infectiousness of arbovirus to mosquitoes. We confirmed this using live hosts. A series of mosquito feedings with *Culex pipiens* was conducted on WNV-infected American robins and common grackles over a range of viremias. Mosquitoes were assayed individually by plaque assay for WNV at 3 to 7 days after feeding. At equivalent viremia, robins always infected more mosquitoes than did grackles. We conclude that the infectiousness of viremic birds cannot always be deduced from viremia alone. If information concerning the infectiousness of a particular bird species is important, such information is best acquired by feeding mosquitoes directly on experimentally infected individuals of that species.

**Data Availability Statement:** All relevant data are within the manuscript and its Supporting information files.

## Author summary

We injected West Nile virus into seronegative individuals of two bird species, American Robin and Common Grackle, and then fed *Culex pipiens* mosquitoes on these birds during the viremic period. We found that, despite having equivalent or lower levels of viremia, robins infected significantly more mosquitoes than did grackles. The reasons for this are not known, but our results indicate that the relationship between host viremia and infectiousness is not always straight forward and may vary among different host species. To gain precise information for a mosquito-borne arbovirus system, it is best to conduct mosquito feedings on live viremic hosts rather than extrapolate the infectiousness of a host species based solely on its viremia profile.

**Funding:** This study was funded by a grant from the National Institute of Allergy & Infectious Diseases (https://www.niaid.nih.gov, Award R21 AI105662 to J.A.V.) The funders had no role in study design, data collection and analysis, decision to publish, or preparation of the manuscript.

**Competing interests:** The authors have declared that no competing interests exist.

## Introduction

Mosquito-borne arboviruses remain a significant cause of human and animal disease [1, 2]. These diseases are maintained principally via horizontal transmission of virus from a viremic host to a susceptible mosquito vector and back again to an immunologically naïve host [3–5]. There can be many factors that influence the efficiency of mosquito arbovirus transmission—perhaps none so important as the quantity of virus ingested by mosquitoes, which in turn is determined by the level and duration of viremia in the host [4]. Other modulating factors may include vector factors (*e.g.*, vector competence [6], feeding preferences [7]), environmental factors (*e.g.*, temperature [8, 9]), and the co-occurrence of microorganisms within the mosquito [10–12] or within the blood of the vertebrate host [13–20].

As part of a study to investigate if pre-existing microfilarial blood parasites could affect the virus infection process in mosquitoes, we examined two passerine bird species; American Robin (*Turdus migratorius*) and Common Grackle (*Quiscalus quiscula*)—both natural hosts for West Nile virus (WNV) and commonly infected with their own distinct species of microfilarial parasites [21]. Unexpectedly, it became apparent that the relationship between host viremia and resultant mosquito infection was less influenced by whether a bird had blood parasites than by what species it belonged to. The theoretical relationship between host viremia and mosquito infection is assumed to be a sigmoidal expression whose shape is largely defined by the level of susceptibility that different mosquito species have to different arboviruses [22]. The host species that produces the viremia is generally assumed to play no role in the virus-mosquito relationship. In this report, we challenge this assumption by presenting a re-analysis of our previous study [21] in which we specifically focus on the host viremia-mosquito relationship for WNV, as observed in two different passerine bird species.

## Materials and methods

### Ethics statement

Research was conducted under Institutional Animal Care and Welfare Committee approved protocols from the University of North Dakota (#1304–2) and the United States Army Medical Research Institute of Infectious Diseases in compliance with the Animal Welfare Act and other Federal regulations relating to the experimental use of vertebrate animals. Collection, transport, and experimentation with migratory birds were conducted under the approval of U. S. Fish & Wildlife Service scientific collection permit MB072162 and collecting permits from the state wildlife agencies of North Dakota and Minnesota.

### Birds

American robins were captured from Roseau and Pennington Co., MN, using mist nets. Common grackles were captured from Grand Forks Co., ND, using baited ground traps. Birds were maintained in outdoor and indoor aviaries at the University of North Dakota where they were screened for blood parasites and antibodies to WNV [23–25]. Preliminary trials were performed to determine appropriate anesthesia doses. Robins (70-90gm) were generally smaller than grackles (80-125gm) but required a higher anesthetic dose (5 mg ketamine + 0.2 mg xylaxine per 100gm BW) than grackles (2 mg ketamine + 0.4 mg xylaxine per 100gm BW) to achieve sufficient level of anesthesia necessary to obtain successful mosquito feedings. Six robins and nine grackles were selected for use and transported during three separate trips in modified pet carriers via commercial airline from Grand Forks, ND, to Washington, D.C., and via automobile to the United States Army Medical Research Institute of Infectious Diseases (USAMRIID) in Frederick, MD. Birds were housed in standard bird cages for an

acclimatization period of $\geq$ 24 h prior to infection with WNV. Because blood parasitemia is a natural condition for these bird species, parasitemias were recorded but birds were not treated with anti-helminthic or anti-protozoal drugs prior to use. The inclusion criterion was simply testing negative to WNV antibodies. However, after experiments were complete, data from one of the nine original grackles (Grackle #42) had to be excluded from this report because it was discovered that a pre-existing microfilarial infection in this bird significantly increased WNV infectivity and dissemination in mosquitoes that fed on this bird [21]. Pre-existing microfilarial infections in four of the other grackles did not affect the course of WNV infectivity in mosquitoes, compared to mosquitoes fed on non-microfilaremic grackles. Similarly, blood parasites (*i.e.*, *Plasmodium*, trypanosomes, microfilariae) in the robins did not affect WNV infectivity to mosquitoes [21].

## Mosquitoes

Two strains of *Cx. pipiens* were used. The 'Rutgers' strain originated from larvae collected in the 1980's at East Brunswick, NJ (F>200). The 'Area B' strain originated from larvae collected in 2010 at Fort Detrick, MD (F~30).

## Virus

A Vero (African green monkey cell) passage 1 of the 397–99 strain of WNV was used throughout this study. This strain originated from an infected crow in 1999 at the Bronx Zoo [26].

## Mosquito infections

Birds were inoculated subcutaneously with 0.2 ml of WNV (6.1 $\log_{10}$ PFU/bird) between 1700 and 1930 h. At intervals of 30, 54 and 78 h after WNV infection, birds were anesthetized and placed through the cotton stockinette of cylindrical screen-topped cardboard cages containing *ca.* 50 to 75 mosquitoes. Mosquitoes were allowed to feed directly on the birds for *ca.* 30 minutes. All unfed and partially fed mosquitoes were removed. Two fully engorged mosquitoes from each cage were removed, triturated, and tested on Vero cells by plaque assay [26] to determine the viremia at the time of feeding. Viral titers per engorged mosquito were converted to $\log_{10}$ PFU/ mL of avian blood (see Table A in S1 Text, for details). Mosquitoes were maintained at 26˚ C and subsamples of 10 to 63 mosquitoes were removed from cages 3, 4, 5, or 7 days later, triturated individually in 1 ml of diluent, and stored at –80˚C until tested for virus by plaque assay.

## Data analyses

The proportions of mosquitoes infected were compared by chi square analyses with Yates correction (Statistix, Tallahassee, FL) among groups of mosquitoes that fed on birds with equivalent viremia. Mixed model logistic regression was used to analyze the probability of mosquitoes becoming infected as a function of host viremia, bird species, and mosquito strain as main fixed effects and possible interactions between these main effects. Individual birds were included in the models as a random effect to account for potential bird-to-bird variation. Each mosquito was represented as a binary value (0 = uninfected; 1 = infected). Logistic regressions were performed using the software, R (version 4.2.1, R Core Team, 2022), examining all main effects and all possible interactions. Initial analysis indicated that 3-way interaction had no significant effect. Therefore, we included the three main effects plus the 2-way interactions of viremia*mosquito strain and viremia*bird species into the final model. Separately, we used probit regression to calculate for each bird species the theoretical concentration of WNV that would infect 50% of the mosquitoes (*i.e.*, the $IC_{50}$). We set $\log_{10}$ viremia as the independent

variable and the proportion of mosquitoes infected at each species-specific viremia (transformed to probits) as the dependent variable (IBM SPSS Statistics, Armonk NY). For comparative purposes, we also included in our probit analysis the previously published dose-response data for *C. pipiens* feeding on WNV-infected baby chickens [27–29]. Comparative $IC_{50}$ values and probit regression slopes of the bird species were considered significantly different from one another if the 95% confidence intervals of respective $IC_{50}$ and slopes did not overlap. To calculate an index of host competence for each bird species, the proportion of mosquitoes that were infected by viremic birds were averaged for each day of the viremic period (*i.e*, 3 days for grackles, 2 days for robins). Daily averages were then summed to yield a total value for overall host competence. These calculations were performed twice—one used the empirically-derived results of mosquito feedings on viremic grackles and robins; the other used the same series of viremias but computed the proportion of infected mosquitoes expected at each viremia, based on the dose-response equation for *C. pipiens* feeding on WNV-infected baby chickens [26–28] (see Tables C, D and E in S1 Text, for details). This was done to evaluate the suitability of substituting the dose-response of baby chickens for those of robins and grackles, as a practical method to determine the relative host competence for these two passerine species based only on their viremia profiles and thereby remove the necessity for conducting mosquito feedings.

## Results

### Viremia profiles

The course and magnitude of WNV viremia for robins and grackles were similar [21] (see also Table A in S1 Text). Peak viremia in six robins occurred 1 to 2 days after inoculation and ranged from 6.5 to 7.4 $\log_{10}$ PFU/mL (average 7.1 ± 0.3). Similarly, peak viremias in eight grackles occurred on days 1 to 2 after inoculation but their magnitudes were more variable, ranging from 6.3 to 8.3 $\log_{10}$ PFU/mL (average 6.9 ± 0.8). Viremia in both species waned or subsided altogether by Day 3.

### Effect of mosquito strain

A total of 1,205 *Cx. pipiens* mosquitoes were tested by plaque assay for WNV infection (see Table B in S1 Text, for details). Of these, 378 fed on robins (187 Area B strain, 191 Rutgers strain), and 827 fed on grackles (469 Area B strain, 358 Rutgers strain). To determine if the two mosquito strains differed in susceptibility, the proportions of mosquitoes infected were compared where the two strains fed concurrently on the same bird or on a bird of the same species with similar viremia. In instances where mosquito strains fed on two con-specific birds having the same viremia, mosquito infection data for respective mosquito strains were combined. The Area B strain was more susceptible to infection than the Rutgers strain, but the magnitude of difference was influenced by bird species (Table 1). For robins, there were seven levels of viremia in which both mosquito strains fed concurrently. Only one (6.7 $\log_{10}$ PFU/ mL) resulted in a significant difference between mosquito strains (p = 0.002). For grackles, there were four levels of viremia in which both mosquito strains fed concurrently. The Area B strain was significantly more susceptible than Rutgers strain in two of the four concurrent feedings (6.2 and 6.3 $\log_{10}$ PFU/mL, p = 0.007 and p = 0.046, respectively) and nearly statistically significant (p = 0.07) in a third (6.4 $\log_{10}$ PFU/mL).

### Comparative infectivity between bird species

To examine comparative infection in mosquitoes fed on the two bird species, the percentages of infected mosquitoes fed on robins were paired with infection percentages of the same

**Table 1. West Nile virus infection rates in two strains of *Culex pipiens* mosquitoes 5 to 7 days after feeding concurrently on birds with the same viremia.** Mosquito infection data are pooled in instances where two birds had the same viremia.

| Bird Species | Bird(s) ID # | Host Viremia (log$_{10}$ PFU/mL) | % Mosquito Infected (number tested) | | p-value |
|---|---|---|---|---|---|
| | | | Area B strain | Rutgers strain | |
| American Robin | 67 | 4.5 | 8% (13) | 10% (10) | 0.692 |
| | 67 | 6.5 | 100% (13) | 90% (31) | 0.612 |
| | 3, 89 | 6.7 | 100% (53) | 77% (48) | 0.002 |
| | 108 | 7.0 | 100% (11) | 100% (23) | 1.000 |
| | 52 | 7.2 | 100% (20) | 100% (20) | 1.000 |
| | 89 | 7.3 | 89% (28) | 91% (35) | 1.000 |
| | 55 | 7.4 | 100% (38) | 100% (14) | 1.000 |
| Common Grackle | 152 | 5.7 | 40% (20) | 27% (11) | 0.479 |
| | 1 | 6.2 | 53% (15) | 9% (23) | 0.007 |
| | 1, 53 | 6.3 | 54% (93) | 32% (32) | 0.046 |
| | 53, 29 | 6.4 | 47% (88) | 21% (19) | 0.074 |

mosquito strain fed on grackles, where the viremias were similar at the time of mosquito feeding (Table 2). There were two exact matches (using Area B strain) where individual birds of both species had matching viremias of 6.5 and 6.7 log$_{10}$ PFU/mL (see pairings 1 & 2, Table 2). In both pairings, viremic robins infected significantly greater proportions of mosquitoes than did grackles with matching viremia (p<0.0001). Three other pairings were examined wherein mosquitoes of the same strain fed on viremic grackles that had four to 10-times more virus than robins (see pairings 3, 4 & 5, Table 2). In each of these pairings, the viremic robins infected significantly greater proportions of mosquitoes than did the viremic grackles (p<0.01), despite the mosquitoes ingesting less WNV from the robins than from the grackles.

## Viremia-Infectivity relationship

Mixed model logistic regression analysis indicated that host viremia had the strongest effect (P<0.00001) on determining the probability of mosquitoes becoming infected (Fig 1). There was also a statistically significant effect due to the interaction between host viremia and bird species (p<0.005), indicating that the probability of mosquito infection increased with increasing viremia differently according to the bird species. This was reflected in the different shapes

**Table 2. West Nile-infected American robins were more infectious to *Culex pipiens* mosquitoes (both strains) than were common grackles with equivalent or higher viremias.**

| Pairing | Mosquito Strain | Bird Species | No. Birds | Viremias (log$_{10}$ PFU / mL) | % Mosquito Infected (number tested) | p-value |
|---|---|---|---|---|---|---|
| 1 | Area B | Robin | 1 | 6.5 | 100% (13) | <0.0001 |
| | | Grackle | 1 | 6.5 | 32% (60) | |
| 2 | Area B | Robin | 2 | 6.7 | 100% (53) | <0.0001 |
| | | Grackle | 2 | 6.7 | 49% (103) | |
| 3 | Area B | Robin | 3 | 7.3–7.4 | 95% (77) | <0.0001 |
| | | Grackle | 1 | 8.0 | 63% (60) | |
| 4 | Rutgers | Robin | 2 | 6.7 | 77% (48) | 0.0058 |
| | | Grackle | 1 | 7.7 | 48% (50) | |
| 5 | Rutgers | Robin | 1 | 7.3 | 91% (35) | 0.0087 |
| | | Grackle | 1 | 8.3 | 65% (63) | |

|  | Estimate | Standard Error | Z value | Pr (>\|z\|) |
|---|---|---|---|---|
| Viremia | 1.173 | 0.254 | 4.624 | **0.000004** |
| Bird Species-Robin | -4.313 | 2.722 | -1.585 | 0.1130 |
| MosqStrain-Rutgers | -2.045 | 1.688 | -1.212 | 0.2257 |
| Viremia*MosqStrain-Rutgers | 0.158 | 0.249 | 0.632 | 0.5272 |
| Viremia*Species-Robin | 1.133 | 0.403 | 2.813 | **0.0049** |

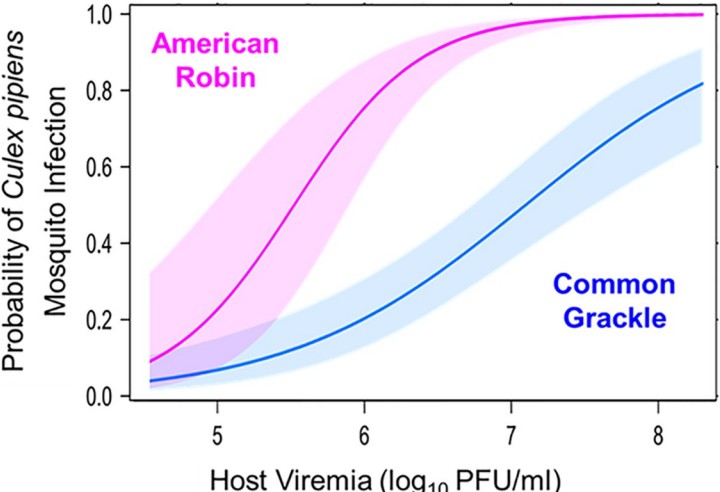

**Fig 1. Relationship between West Nile virus viremia in American robins versus common grackles and infectiousness to *Culex pipiens* mosquitoes, as modeled by logistic regression including a random effects term for individual birds.** Shaded areas in plot represent 95% confidence intervals.

of the dose-response curves from mosquitoes that fed on robins versus grackles (Fig 1). The dose-response for mosquitoes feeding on robins exhibited a steep exponential rise in the infection response starting at *ca*. $\log_{10}$ 5 PFU/mL whereas mosquitoes feeding on grackles exhibited a much shallower rise in the infection response starting at *ca*. $\log_{10}$ 6 PFU/mL).

Probit regression analysis indicated that the levels of host viremia needed to infect 50% of *Cx. pipiens* mosquitoes (*i.e.*, the $IC_{50}$'s) were statistically equivalent between baby chickens (5.7 $\log_{10}$ PFU/mL) and robins (5.6 $\log_{10}$ PFU/mL) and both had overlapping 95% confidence intervals. The $IC_{50}$ for mosquitoes fed on grackles (7.4 $\log_{10}$ PFU/mL) was 50-fold higher and the confidence intervals did not overlap, indicating there was a statistically significant difference in $IC_{50}$ values between mosquitoes fed on grackles versus the other two bird species. Similarly, the slope (*b*) of the dose-response for mosquitoes fed on viremic grackles (*b* = 0.4) was significantly less (*i.e.*, non-overlapping confidence intervals) than for mosquitoes fed on either viremic robins (*b* = 1.1) or viremic baby chickens (*b* = 1.1) (Fig 2).

## Host competence

Index values for host competence were computed separately for robins and grackles based on 1) extrapolations from a regression equation derived from feeding mosquitoes on baby chickens, and 2) actual results from feeding mosquitoes on viremic robins and grackles. Higher values denote higher relative host competence. For robins, the two methods yielded comparable index values. For grackles, the index value based on actual infection data was significantly less

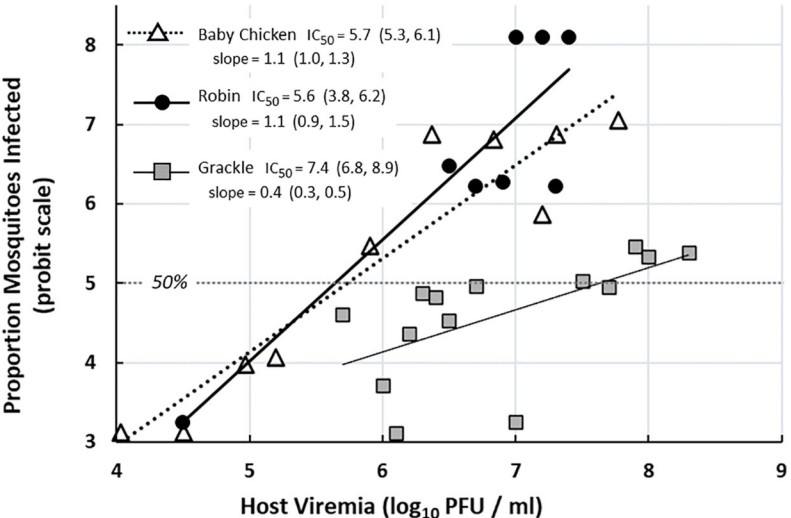

**Fig 2. Dose-response (log-probit scale) of *Culex pipiens* mosquitoes to ingested West Nile virus when fed on viremic baby chickens, American robins, and common grackles.** The $IC_{50}$ (95% confidence) values represent the predicted host viremia ($log_{10}$ PFU/mL) that infects 50% of the mosquitoes. Data for baby chickens were taken from references 26–28.

than the value derived mathematically from the dose-response regression equation of baby chicken viremias (non-overlapping 95% confidence intervals). Moreover, estimates based on the baby chicken regression ranked grackles as having a significantly higher host competence index value than robins while calculations based on actual infection data from viremic robins and grackles ranked robins as having a significantly higher host competence (Table 3).

## Discussion

American robins and common grackles experienced similar levels of viremia when injected subcutaneously with WNV, but robins infected significantly greater percentages of *Cx. pipiens* mosquitoes than grackles—even when robins had substantially lower viremia than grackles (Table 2). At first glance, this seemed remarkable, but in fact, it should not have been totally unforeseen. A number of previous studies have demonstrated that the source and condition of blood in which arboviruses are presented to mosquitoes can significantly affect the efficiency of mosquito infection. Most of these studies utilized *in vitro* mosquito feeding systems—*e.g.*, blood-soaked gauze, membrane feeders, *etc*. The advantages of these systems are that the virus concentration and diluent composition fed to mosquitoes can be precisely controlled and that live animals are not needed for the studies. Using blood-soaked gauze, Hardy and Reeves [29]

**Table 3. Comparative index values (95% confidence intervals) of host competence for West Nile virus-infectiousness to *Culex pipiens* mosquitoes by viremic robins and grackles, as estimated mathematically from a dose-response equation generated from mosquito feedings on viremic baby chickens versus measuring actual mosquito infections that resulted from feedings on viremic robins and grackles.**

| Bird Species | Estimates (95% CI) derived from baby chicken infections | Estimates (95% CI) based on actual infection results |
|---|---|---|
| American Robin | 1.56 (1.38, 1.75) | 1.66 (1.49, 1.83) |
| Common Grackle | 2.36 (2.28, 2.45) | 1.17 (1.06, 1.27) |

 

first reported that Saint Louis encephalitis virus was more infectious to *Cx. tarsalis* mosquitoes when presented in defibrinated blood obtained directly from a viremic chicken than was the same amount of virus obtained from suckling mouse brains and mixed into normal, defibrinated rabbit or chicken blood. Similarly, Huang *et al.* [30] reported that Zika virus mixed in defibrinated whole sheep blood was significantly more infectious to *Aedes aegypti* mosquitoes (64% infected, n = 82) than the same concentration of Zika virus mixed in bovine serum albumin (27% infected, n = 82). They suggested that cellular components of the blood meal may be important in the virus infection process. More recently, Abbo *et al.* [31] reported that Usutu virus mixed in whole human blood (anticoagulation method not stated) was significantly more infectious to two biotypes of *Cx. pipiens* (66% infected, n = 234) than the same concentration of Usutu virus mixed in whole chicken blood (49%, n = 283). In addition to cellular components, there may be soluble factors involved with the virus infection process. The most detailed study describing this phenomenon was that of Marchi *et al.* [32]. They examined the infectivity of African horse sickness virus (Reoviridae) to a midge vector, *Culicoides variipennis*. Infection rates in midges fed 5 days earlier on virus at 7.7 $\log_{10}$ tissue culture infectious dose 50% per ml differed markedly depending on the blood source to which the virus had been added. When added to heparinized horse or dog blood, the virus infected 51% (n = 933) and 41% (n = 209) of the midges respectively, but when added to heparinized sheep or cow blood, the same concentration of virus infected only 26% (n = 160) and 11% (n = 160) of the midges respectively. When purified virus was incubated overnight in horse or dog sera, gel electrophoresis demonstrated that the outer capsid protein VP2 of the virus was cleaved, whereas the VP2 protein remained intact after overnight incubation with cow or sheep sera. Furthermore, pretreatment of virus with chymotrypsin not only cleaved the viral capsid VP2 but when mixed with cow serum and fed to midges, increased the infectivity of cow/virus bloodmeals to midges from 11% (n = 160) to 51% (n = 60). This suggested that in the case of African horse sickness virus and midge vectors, species-specific serum proteases play an important role in determining virus infectivity to the vector.

*In vitro* feeding systems have clear advantages but also have the disadvantage that they require defibrination or addition of anti-clotting agents to the blood. Without normal blood clotting, ingested virions remain distributed randomly throughout the blood bolus instead of being pushed outwards towards the periphery as the result of normal fibrin formation [33]. Concentration of virions on the peripheral of a blood meal during clotting allows more virions to contact receptor sites on the midgut lumen. Not surprisingly, use of live viremic hosts typically results in significantly increased mosquito infection rates compared to *in vitro* feeding systems, even when the same source of blood was used [34–36]. Live host feedings more closely represent the natural situation but can be more challenging because viremias in a live host cannot be controlled. Comparing arboviral infectivity of different host species using live host feedings often require mosquito feedings on multiple animals and/or multiple times throughout the course of viremia, in the hope of obtaining matching viremias in two host species at the time of mosquito feeding. Few studies have done this. As part of a larger study, Reisen *et al.* [37] used restrained, viremic birds to infect mosquitoes with WNV. When two panmictic populations of *Cx. tarsalis* from Kern Co., CA, fed on a house finch (*Carpodacus mexicanus*) with a titer of WNV of 5.4 $\log_{10}$ plaque-forming units (PFU)/mL, and similarly on a 7-day old chicken (*Gallus domesticus*) with a slightly higher titer of WNV of 5.7 $\log_{10}$ PFU/mL, the resulting mosquito infection rate in mosquitoes fed on the viremic finch (94%, n = 17) was significantly higher than in mosquitoes fed on the viremic chick (13%, n = 16) (p<0.00001), despite the viremia in the chick being higher than in the finch. Although this represents only a single comparative feeding, their experiment (together with *in vitro* feeding studies cited

above) suggest that viremic blood in different bird species can significantly alter the infectivity of WNV to *Culex* mosquitoes feeding on viremic birds.

The mechanism(s) responsible for different infectiousness of robins versus grackles is unknown, but it appears unlikely that the increased infectiousness of robins resulted from co-infection of blood parasites [21]. More likely, different infectiousness of the two bird species resulted from innate differences in their blood and how the viremic blood behaved once inside the mosquito midgut. There are at least two possibilities, neither mutually exclusive. First, the WNV virions may be rendered less infective in grackle sera due to the activity (or lack thereof) of species-specific serum proteases, similar to that described for African horse sickness virus in dog/horse versus sheep/cow sera [32]. Alternatively, the kinetics and/or intensity of fibrin formation in the blood of viremic grackles may be altered such that ingested WNV virions are not pushed to the periphery of blood meals within engorged *Cx. pipiens* mosquitoes to the same degree as in robin blood. Reduced localization of virus to the periphery of blood meals would lead to a reduced number of contacts between ingested virions and midgut epithelium, resulting in lowered mosquito infection rates [33]. The notion that the viremic state can alter blood meal clotting is supported by the observation that dengue virus (also a flavivirus) alters the kinetics and intensity of fibrin formation within human sera of dengue patients [38].

Regardless of the mechanism involved, our results complicate the concept of vertebrate host competence, defined as the relative infectiousness of a vertebrate host species to mosquitoes. Originally developed for eastern equine encephalitis virus in European starlings (*Sturnus vulgaris*) with *Culiseta melanura* mosquitoes, Komar *et al.* [39] proposed a simple calculation to derive a numerical value for host competence—the host competence index ($C_i$)—which is the product of the average daily host infectiousness (*i*) to mosquitoes and its duration in days (*d*). This index has become a critical component in estimating relative contributions of different host species in arbovirus transmission cycles in nature [40] and in scenario planning for predicting future invasive arbovirus transmission yet to occur [41].

For the index to be useful, the average daily infectiousness component (*i*) of the index should be determined experimentally. This is done by feeding mosquitoes on a range of virus concentrations and subsequently determining the proportion of mosquitoes infected at each concentration. The results are then used to construct a dose-response equation, making it possible to estimate the proportion of infected mosquitoes generated at different host viremias. Importantly, the viremia/infectiousness relationship must be determined experimentally by feeding mosquitoes on live viremic hosts, not *in vitro* methods (*e.g.*, membrane feeders) because the use of defibrination or anticoagulants in artificial blood meals invariably reduce mosquito infectivity [34–36] and will lead to systemic underestimation of host competency index values.

Host competency index values for WNV in *Cx. pipiens* were calculated and compared among 53 vertebrate species by Kilpatrick *et al.* [42]. In this case, the infectiousness component (*i*) of the index was based on the viremia-infectiousness relationship of *Cx. pipiens* feeding on viremic 1-to-5-day old chickens [26, 27]. No data from mosquito feedings on other bird species were available. Using the viremic profiles of the host species and the viremia-infectiousness equation based on baby chickens, Kilpatrick *et al.* [42] estimated the host competence index for the Common Grackle ($C_{\text{grackle}} \sim 1.7$) to be substantially higher than for the American Robin ($C_{\text{robin}} \sim 1.1$). Although there were some slight differences in the regression equations and calculation used, this is essentially the same thing we observed when host infectiousness to mosquitoes was extrapolated from the baby chicken dose-response equation (Table 3). But when we used actual infection data from viremic robins and grackles, the relative ranking of these two bird species was reversed (Table 3). Clearly, when baby chickens were used as a

surrogate to define the viremia-infectiousness relationship, grackles were over-rated as competent hosts for WNV.

Mosquito feedings on viremic live birds will establish more accurate host competence index values for WNV. But these types of experimental infections present challenges in terms of logistics, securing federal and state collecting permits, determining appropriate anesthesia or restraining methods, addressing compliance/biosafety issues, *etc*. From a practical standpoint, this level of stringency is more realistically done for what may be considered high-priority bird species—*i.e.*, birds known to produce high viremia (*e.g.*, crows, jays), or are numerous and associated with humans (*e.g.*, house sparrows), or are preferentially fed on by vector mosquitoes (*e.g.*, robins). Experimental mosquito feeding on viremic live birds may be less warranted for species that produce low viremia $\leq 4.6 \log_{10}$ PFU/mL (*e.g.*, pigeons, adult chickens) or are uncommon. In all cases, it is important to measure host viremia in terms of infectious virions (*e.g.*, plaque assay), not viral RNA (*via* reverse transcriptase PCR) or protein (*via* immunoassay), if the study objective is to determine the viremia-mosquito infection relationship.

With regard to WNV transmission, the American Robin is now recognized as one of the most important amplifying species in eastern North America due to the observation that *Cx. pipiens* mosquitoes tend to feed preferentially on robins—especially during the early part of the transmission season [7, 43–45]. The present study demonstrates that viremic robins are also more innately infectious to *Cx. pipiens* than are viremic grackles, further underscoring the importance of the American Robin as a key amplifying host for WNV. More generally, our studies demonstrate that for WNV (and possibly other arboviruses), the relationship between host viremia and the efficiency by which a viremic host infects mosquitoes is not fixed and cannot be determined solely by measuring host viremia. The relationship may also depend on the host species involved.

## Supporting information

**S1 Text.** Table A. Information on birds and viremias as determined by plaque assay of engorged mosquitoes frozen immediately after feeding on individual birds. Table B. Host viremia, blood parasite status, and % mosquitoes infected. Table C. Published sources used to generate the host viremia:mosquito infection relationship for baby chickens and *Culex pipiens* mosquitoes. Table D. Method used to compute comparative host competence index values. Table E. Method used to compute comparative host competence index values.
(DOCX)

## Acknowledgments

Students in the Vaughan laboratory helped capture and screen birds. Ms. Danielle Kvasager and Mr. Chad Stromlund captured American robins and screened them for microfilariae and trypanosomes. Ms. Sarina Bauer assisted with the molecular diagnosis of hemosporidian parasites. Mr. Lei Guo assisted with immunoassays to screen birds for antibodies to WNV. Technicians and research fellows in the Turell laboratory assisted with experimental procedures. Dr. Elizabeth Andrews and Ms. Juanita Hinson conducted mosquito trituration and plaque assays. Ms. Denise Nash helped rear the mosquitoes used in this study.

## Disclaimers

Opinions, interpretations, conclusions, and recommendations expressed in this article are those of the authors and do not reflect the official policy of the Department of Army, Department of Defense, or the U.S. Government.

## Author Contributions

**Conceptualization:** Jefferson A. Vaughan, Michael J. Turell.

**Data curation:** Jefferson A. Vaughan, Michael J. Turell.

**Formal analysis:** Jefferson A. Vaughan, Robert A. Newman.

**Funding acquisition:** Jefferson A. Vaughan.

**Investigation:** Jefferson A. Vaughan, Michael J. Turell.

**Methodology:** Jefferson A. Vaughan, Michael J. Turell.

**Project administration:** Michael J. Turell.

**Resources:** Jefferson A. Vaughan, Michael J. Turell.

**Writing – original draft:** Jefferson A. Vaughan.

**Writing – review & editing:** Jefferson A. Vaughan, Robert A. Newman, Michael J. Turell.

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
