## [Decision Letter · Decision Letter 0]

24 Jun 2022

Dear Dr. Vaughan,

Thank you very much for submitting your manuscript "Bird species define the relationship between West Nile viremia and infectiousness to Culex pipiens mosquitoes" for consideration at PLOS Neglected Tropical Diseases. As with all papers reviewed by the journal, your manuscript was reviewed by members of the editorial board and by several independent reviewers. In light of the reviews (below this email), we would like to invite the resubmission of a significantly-revised version that takes into account the reviewers' comments. 

We cannot make any decision about publication until we have seen the revised manuscript and your response to the reviewers' comments. Your revised manuscript is also likely to be sent to reviewers for further evaluation.

Sincerely,

Duane J. Gubler

Associate Editor

Scott Weaver

Deputy Editor

Reviewer's Responses to Questions

**Key Review Criteria Required for Acceptance?**

**Methods**

-Are the objectives of the study clearly articulated with a clear testable hypothesis stated?

-Is the study design appropriate to address the stated objectives?

-Is the population clearly described and appropriate for the hypothesis being tested?

-Is the sample size sufficient to ensure adequate power to address the hypothesis being tested?

-Were correct statistical analysis used to support conclusions?

-Are there concerns about ethical or regulatory requirements being met?

Reviewer #1: A few more details could be added to this section:

1. Were the birds tested daily for viremia? How was the duration of infectiousness determined. With HOSP hosts, mosquitoes can be infected for up to 2 wks post infection despite the waning viremia.

2. What was the sex and age of the host birds? 

3. It was unclear why some birds with a microfilaremia affected mosquito infection while others did not. 

4. How were the birds anesthetized? What was used? 

5. Unclear why the now essentially extinct NY99 strain was used in these experiments.

Reviewer #2: (No Response)

Reviewer #3: See attached

**Results**

-Does the analysis presented match the analysis plan?

-Are the results clearly and completely presented?

-Are the figures (Tables, Images) of sufficient quality for clarity?

Reviewer #1: This section was somewhat tedious, but generally well done.

Reviewer #2: (No Response)

Reviewer #3: See attached

**Conclusions**

-Are the conclusions supported by the data presented?

-Are the limitations of analysis clearly described?

-Do the authors discuss how these data can be helpful to advance our understanding of the topic under study?

-Is public health relevance addressed?

Reviewer #1: In general the data were clearly presented, appropriately analyzed and support the conclusions.

Reviewer #2: (No Response)

Reviewer #3: See attached

**Editorial and Data Presentation Modifications?**

Reviewer #1: No comment. See below.

Reviewer #2: (No Response)

Reviewer #3: See attached

**Summary and General Comments**

Reviewer #1: Two colonized strains of Cx. pipiens were used to compare the host competence of American Robins and Common Grackles for the NY99 strain of WNV. Chicks were used as a laboratory model comparison. Data clearly showed that mosquito infection differed among host species, even when fed on birds with comparable viremias. The authors discuss how these kinds of data confound avian host competence estimated strictly from measuring daily viremia levels. The authors acknowledge, however, problems in the complexity of doing these types of experiments using the wide variety of mosquito and host species involved in WNV transmission. 

Minor comments have been inserted on the attached manuscript using tracked changes. A few summary thoughts include:

1. Literature review. The authors have missed quite a few papers where mosquito infection was used to compare avian host competence, making their current study less unique than described in the Introduction. 

2. Introduction. I think too much space is given to a review of papers showing methods of mosquito in vitro infection can alter vector competence results. The focus should be on in vivo information as this pertains to the experiments at hand. 

 2. Conclusions. The dynamics of WNV transmission is complex, as shown in the current paper where there were differences seen between pipiens colonies and host bird species. Therefore, I think that generalizations used for gross comparisons still are useful and I wouldn't throw out the Komar host competence index just yet. As used by Kilpatrick and Wheeler et al. it still provides a useful way of comparing avian species.

Reviewer #2: Vaughn and Turell present a small study with large implications. They demonstrate that two bird species circulating an equivalent viremia infect mosquitoes with differing efficiencies. This is due to innate differences in the blood of the two species, as yet unidentified. This has several important implications. One is that estimating the relative contribution of different host species to arbovirus amplification is more complex than previously believed. Another is that as yet unidentified blood components modulate vector competence. The discovery of these component(s) could yield new tools in the prevention and control of arbovirus transmission.

General Comments

1. Overall, the paper is very well written.

2. Several different types of units are used for describing virus concentration. TCID50, IC50, LC50, pfu, etc. These need to be reviewed and clarified. At a minimum, spell out abbreviations when first used.

3. The capitalized name of a bird species refers to the species as a whole, not individual animals. Therefore, these should not be pluralized. There is only one American Robin, but there can be multiple American robins. Also be consistent with capitalization of bird names. In the Intro, you do not capitalize house finch.

4. In the second paragraph of the introduction, treatment of vertebrate reservoir competence is confusing because you are actually referring to reservoir capacity rather than competence. Competence describes the potential to infect mosquitoes, whereas capacity measures the relative importance of different species as amplifiers. The background on transmission dynamics can probably be streamlined into a single paragraph. If necessary, further description of the processes involved could be added to the discussion.

5. The Intro is too heavy on background information and too light on presenting the specific scientific problem to be addressed and a hypothesis statement. Please add a hypothesis to the final paragraph in the Introduction. 

6. Why were American Robin and Common Grackle chosen for this study? Was there prior evidence that the blood of these two species differed with respect to WNV infectiousness?

7. When presenting average viremia values, do these represent the log of the mean, or the mean of the logs? In other words, do you log transform for presentation purposes before or after calculating the mean value? It makes a difference. This should be explained in the methods.

8. Why are you including the data for the Rutgers mosquito strain?? Do these data add to the quality of the experiment? 

9. Supplemental data file is provided, but I did not notice a citation or reference to these data in the main text.

Specific comments:

Line 138 quiscula is misspelled.

Line 167-169. In order to make your study repeatable by a reader, please explain how you converted three engorged mosquitoes into a viremia measure. Did you measure the volume of blood in each mosquito? Were the mosquitoes each tested in order to produce a mean viremia determination?

Line 169-170. Why were mosquitoes removed for testing on four different days post-feeding? Was there a difference for mosquitoes that incubated for 3 days versus 7 days?

Line 260 Use “index values” rather than “indices”. You are discussing multiple values within a single index. Also there are probably some fancy statistics to generate confidence intervals around the index values, and a p-value for the comparison. I also question the precision of these values. Is reporting out to three significant figures appropriate for any of the measures in this Table?

Line 286 Fig 1 can be gray-scale. Color not required.

Line 302 dog/horse versus cattle/sheep sera.

Line 363. And vector species, for that matter!

Reviewer #3: See attached

PLOS authors have the option to publish the peer review history of their article (what does this mean?). If published, this will include your full peer review and any attached files.

Reviewer #1: No

Reviewer #2: Yes: Nicholas Komar

Reviewer #3: No
---

## [Editor Report · Decision Letter 1]

20 Sep 2022

Dear Dr. Vaughan,

We are pleased to inform you that your manuscript 'Bird species define the relationship between West Nile viremia and infectiousness to Culex pipiens mosquitoes' has been provisionally accepted for publication in PLOS Neglected Tropical Diseases.

Best regards,

Duane J. Gubler

Academic Editor

Scott Weaver

Section Editor

---

## [Editor Report · Acceptance letter]

30 Sep 2022

Dear Dr. Vaughan,

We are delighted to inform you that your manuscript, "Bird species define the relationship between West Nile viremia and infectiousness to *Culex pipiens* mosquitoes," has been formally accepted for publication in PLOS Neglected Tropical Diseases.

Best regards,

Shaden Kamhawi

co-Editor-in-Chief

Paul Brindley

co-Editor-in-Chief
